# Anillin facilitates septin assembly to prevent pathological outfoldings of central nervous system myelin

Michelle S Erwig[1], Julia Patzig[1], Anna M Steyer[1,2,3], Payam Dibaj[1], Mareike Heilmann[4], Ingo Heilmann[4], Ramona B Jung[1], Kathrin Kusch[1], Wiebke Möbius[1,2], Olaf Jahn[5], Klaus-Armin Nave[1], Hauke B Werner[1]*

[1]Department of Neurogenetics, Max Planck Institute of Experimental Medicine, Göttingen, Germany; [2]Electron Microscopy Core Unit, Max Planck Institute of Experimental Medicine, Göttingen, Germany; [3]Center for Nanoscale Microscopy and Molecular Physiology of the Brain, Göttingen, Germany; [4]Department of Cellular Biochemistry, Institute of Biochemistry and Biotechnology, Martin-Luther-University Halle-Wittenberg, Halle, Germany; [5]Proteomics Group, Max Planck Institute of Experimental Medicine, Göttingen, Germany

**Abstract** Myelin serves as an axonal insulator that facilitates rapid nerve conduction along axons. By transmission electron microscopy, a healthy myelin sheath comprises compacted membrane layers spiraling around the cross-sectioned axon. Previously we identified the assembly of septin filaments in the innermost non-compacted myelin layer as one of the latest steps of myelin maturation in the central nervous system (CNS) (Patzig et al., 2016). Here we show that loss of the cytoskeletal adaptor protein anillin (ANLN) from oligodendrocytes disrupts myelin septin assembly, thereby causing the emergence of pathological myelin outfoldings. Since myelin outfoldings are a poorly understood hallmark of myelin disease and brain aging we assessed axon/myelin-units in *Anln*-mutant mice by focused ion beam-scanning electron microscopy (FIB-SEM); myelin outfoldings were three-dimensionally reconstructed as large sheets of multiple compact membrane layers. We suggest that anillin-dependent assembly of septin filaments scaffolds mature myelin sheaths, facilitating rapid nerve conduction in the healthy CNS.
DOI: https://doi.org/10.7554/eLife.43888.001

*For correspondence:
hauke@em.mpg.de

## Introduction

Fast, saltatory nerve impulse conduction in the central nervous system (CNS) of vertebrates is facilitated by the ensheathment of axons with multiple layers of insulating oligodendroglial membrane, termed myelin (*Nave and Werner, 2014*; *Hartline and Colman, 2007*). Myelin compaction along the extracellular membrane surface (intraperiod line) involves cholesterol-associated transmembrane proteins, such as proteolipid protein (PLP) (*Simons et al., 2000*; *Werner et al., 2013*), which exhibits adhesive forces (*Bakhti et al., 2013*; *Bizzozero et al., 2001*) that prevent splitting of myelin lamellae (*Lüders et al., 2017*; *Möbius et al., 2016*; *Rosenbluth et al., 2006*). At the intracellular membrane surface (major dense line), myelin basic protein (MBP) facilitates the tight association of myelin layers by covering the negatively charged headgroups of membrane phospholipids (*Musse et al., 2008*; *Nawaz et al., 2009*; *Nawaz et al., 2013*). The adhesive function of MBP can be counteracted by the presence of cyclic nucleotide phosphodiesterase (CNP), thereby regulating the developmental closure of cytoplasmic channels that flank compacted myelin (*Snaidero et al., 2017*). In mature myelin, CNP is thus largely confined to non-compacted myelin (*Brunner et al., 1989*; *Trapp et al., 1988*). The relevance of PLP, MBP and CNP for the regular ultrastructure of myelin is reflected by their high

abundance in biochemically purified myelin membranes (*Jahn et al., 2009*) and the myelin defects in the corresponding mouse mutants.

In addition to the delamination of single myelin layers, pathological destabilization of myelin was observed as pathological outfoldings of entire stacks of compacted myelin membranes in several myelin mutants, (*Patzig et al., 2016a*) and upon normal brain aging (*Peters, 2002*; *Sturrock, 1976*). We recently found that myelin outfoldings correlate with a loss of septins, which are cytoskeletal proteins of comparatively lower abundance in the myelin proteome (*Patzig et al., 2016a*). Septins are widely expressed and control the rigidity of the membranes they are associated with (*Bridges and Gladfelter, 2015*; *Gilden and Krummel, 2010*). Localized in the non-compacted, adaxonal myelin layer adjacent to the inner-most compacted myelin membrane, myelin septin filaments are assembled from the monomers SEPT2, SEPT4, SEPT7 and SEPT8 in 1:1:2:2 stoichiometry (*Patzig et al., 2016a*). This marks a canonical but distinct composition of subunits when compared with the higher order structures of septins in other cell types (*Barral and Kinoshita, 2008*; *Dolat et al., 2014*). Recently, we proposed that septin filaments provide a scaffold that prevents detachment of the compacted myelin layers from the adaxonal myelin membrane and thus outfoldings of entire myelin sheaths (*Patzig et al., 2016a*).

Septins associate with the pleckstrin homology (PH)-domain containing adaptor protein anillin (ANLN) and its homologs in drosophila embryos (*El Amine et al., 2013*; *Field et al., 2005*; *Liu et al., 2012*), budding yeast (*Eluère et al., 2012*; *Kang et al., 2013*; *Tasto et al., 2003*) and mouse NIH3T3-fibroblasts (*Kinoshita et al., 2002*). Here, the role of anillin in the formation of contractile septin rings is a conserved step of cytokinesis (*Piekny and Maddox, 2010*) and essential for cell division (*Zhang and Maddox, 2010*). However, in post-mitotic cells the interactions between septins and anillin are not understood. In the adult CNS, expression of *Anln* mRNA is highest in myelinating oligodendrocytes when assessed by RNA-Seq (www.web.stanford.edu/group/barres_lab/cgi-bin/igv_cgi_2.py?lname=anln) (*Zhang et al., 2014*), single-cell transcriptomics (www.linnarsson-lab.org/cortex) (*Zeisel et al., 2015*) and in situ-hybridization (mouse.brain-map.org/gene/show/44585) (*Lein et al., 2007*).

Here we show that oligodendroglial anillin serves a crucial function in myelination. Conditional mouse mutants lacking expression of the *Anln*-gene in mature oligodendrocytes fail to assemble septin filaments, display large myelin outfoldings similar to those of *Sept8*-mutant mice (*Patzig et al., 2016a*) and exhibit reduced nerve conduction velocity. This work thus establishes a crucial function for anillin-dependent assembly of myelin septin filaments in scaffolding CNS myelin to enable rapid nerve conduction, thereby demonstrating a vital function of anillin unrelated to cytokinesis.

## Results and discussion

To address a functional connection between anillin and myelin septins, we first asked if the protein is also enriched in CNS myelin. Indeed, anillin was detected in myelin when biochemically purified from mouse brains at P75, while it was virtually undetectable in brain lysates when loading the same amount of protein (*Figure 1A*). This implies that anillin is enriched in myelin similar to SEPT8 variant 1 (SEPT8_v1; according to nomenclature at Ensembl.org) (*Patzig et al., 2016a*) or myelin oligodendrocyte glycoprotein (MOG) (*Linnington et al., 1984*). In comparison, the axonal marker Tubulin-beta3/TUJ1 was diminished in myelin compared to brain lysate (*Figure 1A*). To determine the localization of anillin, we performed immunohistochemistry and confocal microscopy of longitudinal spinal cord sections. We found that anillin-immunolabeling parallels but does not overlap with axonal neurofilament (NF) labeling (*Figure 1B*). Most anillin-immunolabeling was reminiscent of the longitudinal myelin septin filaments (*Figure 1C*) (*Patzig et al., 2016a*).

Considering that the abundance of *Anln* mRNA increases over 10-fold coinciding with the differentiation of oligodendrocyte progenitor cells to myelinating oligodendrocytes (*Zhang et al., 2014*) we tested whether the abundance of ANLN increases with the developmental maturation of myelin. Indeed, by immunoblotting of myelin purified from mouse brains at postnatal day 15 (P15), P18, P21 and P24, the abundance of ANLN increased (*Figure 1D*) similar to that of SEPT2. When immunolabeling ANLN together with SEPT8 on longitudinal optic nerve sections at P15, P21 and P28, first co-

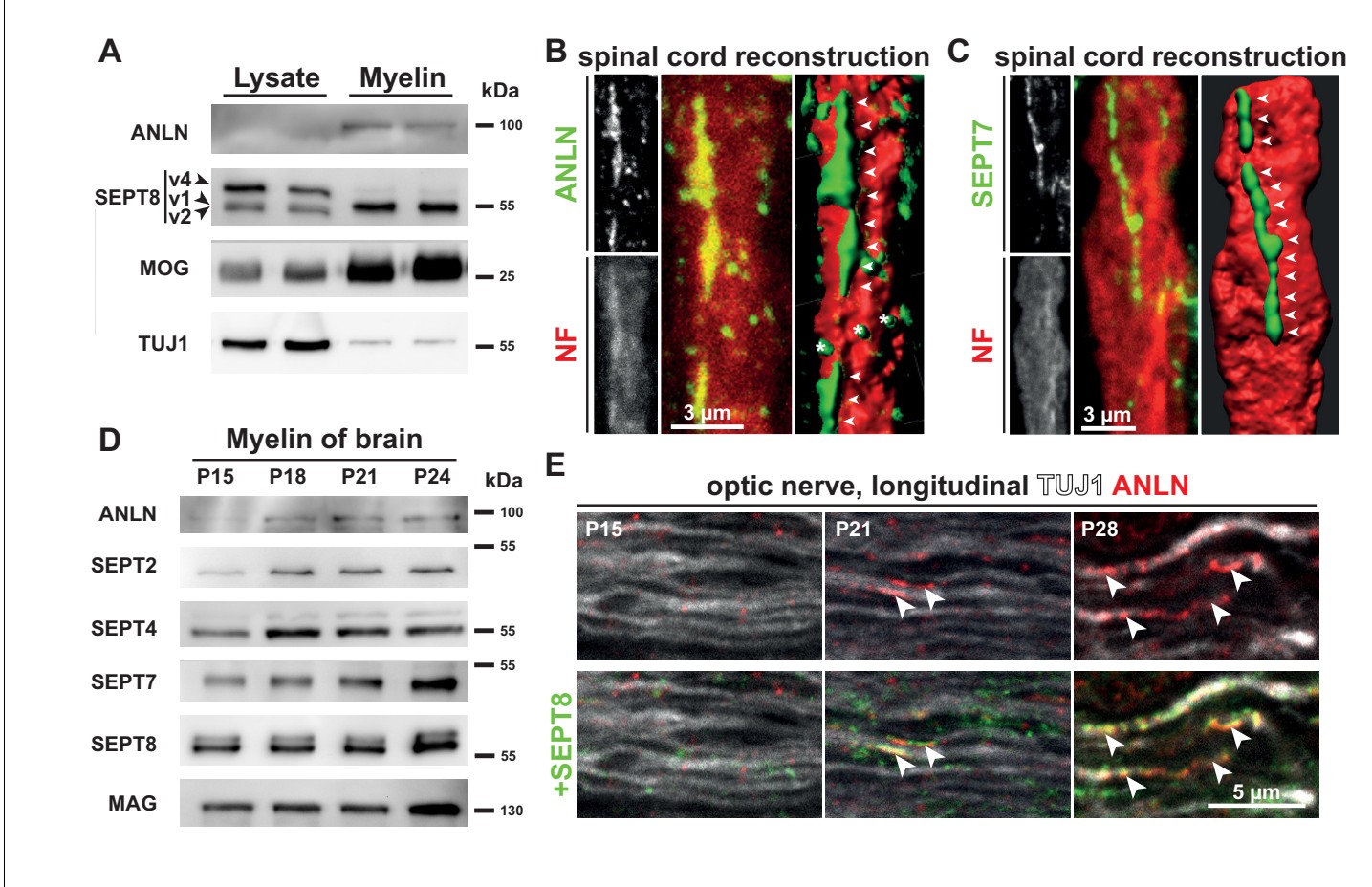

**Figure 1.** Co-distribution of anillin with myelin septins. (A) Immunoblotting of myelin purified from the brains of wild type mice at P75 compared to brain lysates indicates that anillin (ANLN) is enriched in myelin similar to septin 8 variant 1 (SEPT8_v1). The same amount of protein was loaded. The myelin marker MOG and the axonal marker TUJ1 served as controls. Blot shows n = 2 mice per genotype representative of n = 3 mice per genotype. (B–C) Immunofluorescent signal of ANLN (green in B) and SEPT7 (green in C) extends longitudinally along axons identified by neurofilament-labelling (red in B–C). Additional ANLN-immunopositive puncta (asterisks in B) were not evidently associated with filamentous structures (arrowheads in B,C). The panels show maximal projections of confocal stacks and 3-dimensional reconstructions of longitudinally sectioned spinal cord of P75 WT mice. Images representative of three mice. (D) Immunoblotting of myelin purified from the brains of wild-type mice at P15, P18, P21 and P24 indicates that the abundance of ANLN in myelin increases with maturation. Myelin septins (SEPT2, SEPT4, SEPT7, SEPT8) and MAG served as control. Blot shows n = 1 mouse per timepoint. (E) Immunolabelling of longitudinally sectioned WT optic nerves detects ANLN (red) in proximity to SEPT8 (green); co-labeled structures (arrowheads) were seen occasionally at P21 and frequently at P28 but not at P15. TUJ1 served as axonal marker. Images representative of three experiments.

DOI: https://doi.org/10.7554/eLife.43888.002

The following figure supplement is available for figure 1:

**Figure supplement 1.** Co-labeling of ANLN and SEPT8 in various white matter tracts.

DOI: https://doi.org/10.7554/eLife.43888.003

labeled structures were occasionally detected at P21 but frequently seen at P28 (*Figure 1E*). Also in other white matter tracts, ANLN-immunolabeling was largely in proximity to SEPT8-immunolabeling, as seen in the fimbria and the corpus callosum by immunohistochemistry of coronal brain sections from wild type mice at P75 (*Figure 1—figure supplement 1A,B*). Thus, expression of the cytoskeletal adaptor protein ANLN is strongly enriched in mature oligodendrocytes, in which it largely co-distributes with myelin septin filaments that localize to the non-compacted adaxonal myelin layer.

We previously noted that the presence of pathological myelin outfoldings in several myelin mutant mice correlates with reduced abundance of both myelin septins and ANLN (*Patzig et al., 2016a*). However, it remained unknown whether diminishment of ANLN represents a mere

epiphenomenon of septin loss or if ANLN has a function in the assembly of septin filaments. To discriminate between these alternative hypotheses, we generated mouse mutants in which exon 4 of the *Anln* gene is flanked by loxP-sites (*Figure 2—figure supplement 1A*). Appropriate breedings yielded *Anln$^{flox/flox}$;Cnp$^{Cre/WT}$* mice, in which Cre recombinase is expressed under control of the *Cnp* promoter (*Lappe-Siefke et al., 2003*) to mediate recombination (*Figure 2—figure supplement 1B*) in myelinating oligodendrocytes. Conditional mutants and control mice were born at the expected frequencies, and the major white matter regions developed normally as judged by light microscopic visualization of myelin upon silver impregnation (*Figure 2A,A'*). However, electron microscopic analysis revealed the presence of numerous myelin outfoldings in the CNS of *Anln$^{flox/flox}$;Cnp$^{Cre/WT}$* mice (*Figure 2B,C*), very similar to *Sept8$^{null/null}$* and *Sept8$^{flox/flox}$;Cnp$^{Cre/WT}$* mutants (*Patzig et al., 2016a*). Analysis of myelin sheath thickness (*Figure 2D,D'*), the percentage of myelinated axons (*Figure 2E*), degenerating/degenerated axons (*Figure 2F*) and secondary neuropathology (*Figure 2—figure supplement 2*) did not reveal further abnormalities in *Anln$^{flox/flox}$;Cnp$^{Cre/WT}$* mice, implying that myelin outfoldings represent a very specific neuropathology.

To test if myelin outfoldings in *Anln$^{flox/flox}$;Cnp$^{Cre/WT}$* mice impair CNS function in vivo, we measured nerve conduction in the spinal cord at 6 months of age. Indeed, nerve conduction velocity was reduced by 15.5% in *Anln$^{flox/flox}$;Cnp$^{Cre/WT}$* compared to control mice (*Figure 2G*), very similar to *Sept8*-mutants (*Patzig et al., 2016a*). Considering that slowed nerve conduction can be caused by structural changes of the nodes of Ranvier (*Arancibia-Cárcamo et al., 2017*) we performed immunohistochemistry for the nodal and paranodal markers Nav1.6 and CASPR, respectively, and determined the density of the nodes (*Figure 2—figure supplement 3A*) as well as their length and diameter (as indicated in *Figure 2—figure supplement 3B*). This analysis did not reveal any nodal or paranodal abnormality in *Anln$^{flox/flox}$;Cnp$^{Cre/WT}$* mice (*Figure 2—figure supplement 3C–F*). Although we cannot formally rule out other unidentified alterations in the CNS of *Anln$^{flox/flox}$;Cnp$^{Cre/WT}$* mice, myelin outfoldings are the most likely cause of reduced nerve conduction velocity.

At the molecular level we asked whether recombination of the *Anln*-gene in mature oligodendrocytes affects the protein composition of myelin. As expected, ANLN was undetectable by immunoblot analysis of myelin purified from the brains of *Anln$^{flox/flox}$;Cnp$^{Cre/WT}$* mice (*Figure 2—figure supplement 1C*). Interestingly, in these mutants the abundance of SEPT8 in myelin was reduced (*Figure 2—figure supplement 1C*). This prompted us to analyze the entire myelin proteome by quantitative label-free mass spectrometry (*Figure 3—source data 1*). ANLN was readily detectable in myelin purified from the brains of control mice but not identified in *Anln$^{flox/flox}$;Cnp$^{Cre/WT}$* myelin (*Figure 3—figure supplement 1A*). The abundance of all myelin septins (SEPT2, SEPT4, SEPT7, SEPT8) was strongly reduced in myelin purified from the brains of *Anln$^{flox/flox}$;Cnp$^{Cre/WT}$* mice (*Figure 3A,B*, *Figure 3—figure supplement 1B*). Interestingly, the abundance of two GTPases of the Rho-subfamily, CDC42 and RHOB, also appeared reduced, although less than the applied threshold of a 2-fold change (*Figure 3—figure supplement 1B*). Importantly, the abundance of the major myelin marker proteins (*Figure 3—figure supplement 1C*, *Figure 3C*) and cytoskeletal proteins associated with the actin cytoskeleton or microtubules (*Figure 3—figure supplement 1D*) was unaltered.

Since a reduction of the membrane phospholipid phosphatidylinositol (4,5)-bisphosphate (PtdIns(4,5)P$_2$) in myelin of *Pten$^{flox/flox}$;Cnp$^{Cre/WT}$* mice causes a loss of septins and ANLN (*Patzig et al., 2016a*), we asked whether *vice versa* the absence of ANLN from myelin affects PtdIns(4,5)P$_2$–levels. Indeed, by quantitative assessment of label-free lipid extracts (*Goebbels et al., 2010*; *König et al., 2008*) we found a decreased abundance of PtdIns(4,5)P$_2$ in myelin purified from the brains of *Anln$^{flox/flox}$;Cnp$^{Cre/WT}$* compared to control mice (*Figure 3D*).

By qRT-PCR, cDNA-fragments for *Sept2*, *Sept4*, *Sept7*, *Sept8* were amplified with equal efficiency from mutant and control corpus callosi (*Figure 3E*), suggesting that the loss of myelin septins is a posttranscriptional event secondary to ANLN-deficiency. Most likely, myelin septin monomers are degraded if not incorporated into filamentous higher order structures, the formation of which is facilitated by ANLN. The cDNA-fragments for *Rhob* and *Cdc42* were also amplified with equal efficiency from mutant and control corpus callosi (*Figure 3—figure supplement 1E*). Conversely, cDNA fragments for *Anln* were virtually undetectable in *Anln$^{flox/flox}$;Cnp$^{Cre/WT}$* corpus callosi (*Figure 3E*), indicating that expression of *Anln* mRNA in the adult CNS is strongly enriched in mature oligodendrocytes in accordance with RNA-Seq data (*Zeisel et al., 2015*; *Zhang et al., 2014*).

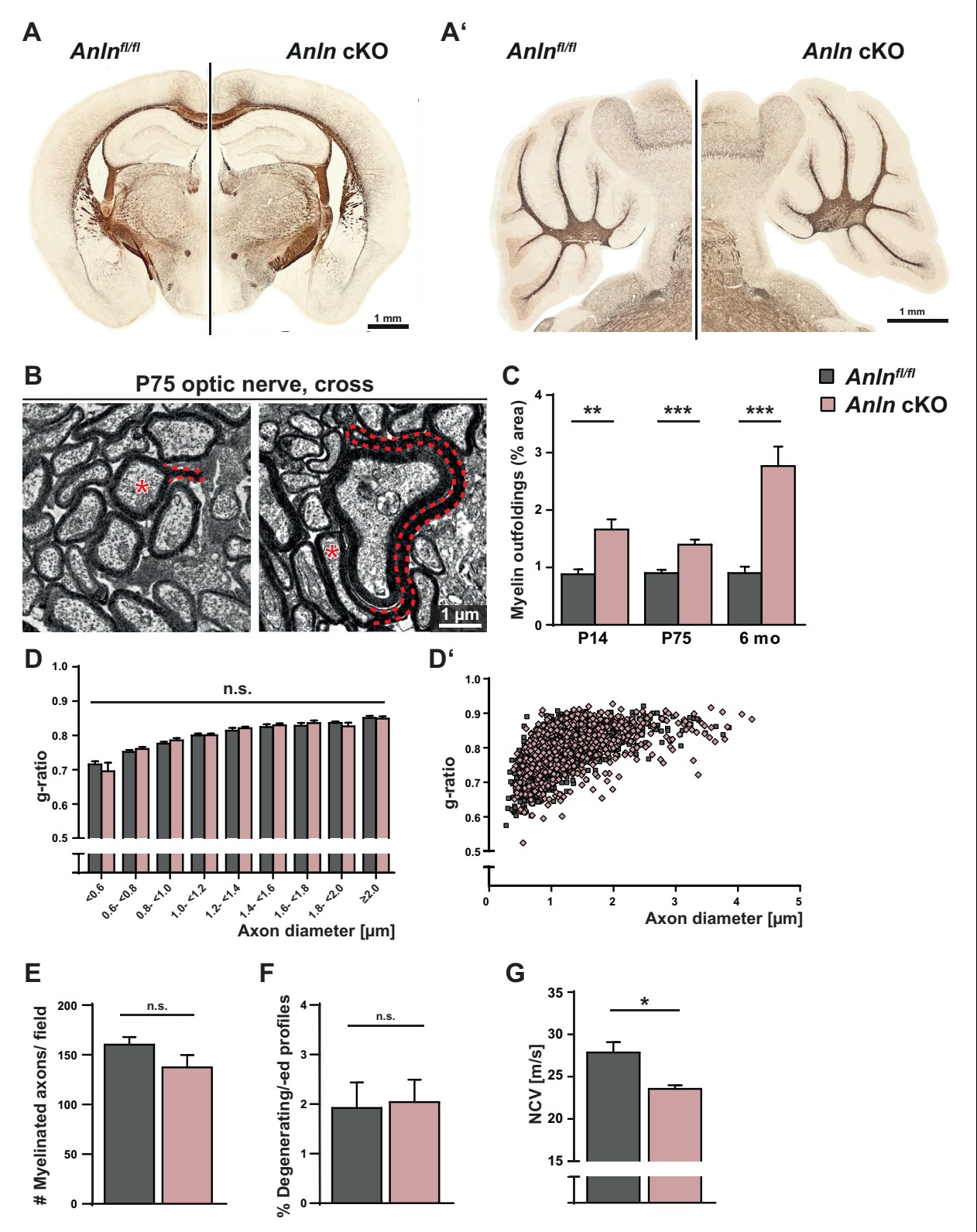

**Figure 2.** Myelin outfoldings and reduced nerve conduction velocity in mice lacking oligodendroglial expression of ANLN. (**A–A'**) Silver impregnation (in brown) visualizes myelinated fiber tracts in mice lacking ANLN from myelinating cells (*Anln*<sup>fl/fl</sup>;*Cnp*<sup>Cre/WT</sup>-mice; *Anln* cKO) and in control mice (*Anln*<sup>fl/fl</sup>) at P75. (**A**) displays coronal brain sections; **A'** shows sagittal sections through the cerebellum. Images representative of three mice per genotype. For generation and validation of *Anln* cKO mice see **Figure 2—figure supplement 1**. (**B**) Electron micrographs of optic nerves exemplify myelin outfoldings
*Figure 2 continued on next page*

*Figure 2 continued*

at P75. Stippled lines highlight myelin outfoldings; associated axons are marked with asterisks. (C) Quantitative evaluation of electron micrographs of optic nerves reveals progressive emergence of myelin outfoldings in adult *Anln^fl/fl;Cnp^Cre/WT* mice (*Anln* cKO). Mean +/SEM. n = 4–6 mice per genotype and age; two-tailed unpaired t-test P14 p=0.0076; P75 p=0.0009; 6mo p=0.0007. (D,D') g-ratio analysis of electron micrographs of optic nerves at six mo indicates normal myelin sheath thickness in *Anln* cKO mice. Mean +/SEM. Not significant according to two-way ANOVA (p=0.9279). (E) Quantitative evaluation of electron micrographs of optic nerves at six mo reveals a normal frequency of myelinated axons in *Anln* cKO mice. Mean +/SEM. n = 4–5 mice per genotype; not significant (n.s.) according to two-tailed unpaired t-test (p=0.1827). (F) Quantitative evaluation of electron micrographs of optic nerves at six mo indicates that there is no increased frequency of degenerating/degenerated axons in *Anln* cKO mice. Mean +/SEM. n = 4–5 mice per genotype; not significant (n.s.) according to two-tailed unpaired t-test (p=0.8664). For immunohistochemical assessment of neuropathology see *Figure 2—figure supplement 2*. (G) Electrophysiological measurement reveals reduced nerve conduction velocity in the spinal cord of *Anln* cKO compared to control (*Anln^fl/fl*) mice at six mo. Mean +/SEM. n = 7–11 mice per genotype; two-tailed unpaired t-test (p=0.0149). For assessment of density and dimensions of the nodes of Ranvier see *Figure 2—figure supplement 3*.

DOI: https://doi.org/10.7554/eLife.43888.004

The following figure supplements are available for figure 2:

**Figure supplement 1.** Generation of mice lacking expression of ANLN from myelinating oligodendrocytes (Anln cKO mice).

DOI: https://doi.org/10.7554/eLife.43888.005

**Figure supplement 2.** ANLN deficiency in oligodendrocytes does not cause axonopathy, astrogliosis or microgliosis.

DOI: https://doi.org/10.7554/eLife.43888.006

**Figure supplement 3.** ANLN deficiency in oligodendrocytes does not cause alterations of density or structure of nodes of Ranvier.

DOI: https://doi.org/10.7554/eLife.43888.007

To determine the morphology of myelin outfoldings three-dimensionally, we performed focused ion beam-scanning electron microscopy (FIB-SEM) of optic nerves dissected from *Anln^flox/flox;Cnp^Cre/WT* and control mice, covering a depth of about 23 μm in each tissue. In control nerves, reconstruction of axonal plasma membranes (false-colored in blue in *Figure 4*) and myelin (false-colored in yellow in *Figure 4*) revealed a largely regular association of myelinated axons with their myelin sheaths (exemplified in *Figure 4A,A'* and *Video 1*). In comparable blocks of optic nerves dissected from *Anln^flox/flox;Cnp^Cre/WT* mice, FIB-SEM allowed the 3D-reconstruction of a number of myelin outfoldings (*Figure 4B',B'''* and *Video 2* and *Video 3*). Most of them were in internodal segments; we observed only a single myelin outfolding close to a node of Ranvier. The observed myelin outfoldings measured between 10 μm and 15 μm in length. Thus, myelin outfoldings do not adopt pin-needle-like shape but represent large sheets of compacted multilayered membrane stacks that extend for considerable distance away from the myelinated axon.

Outfoldings of compact CNS myelin are a neuropathological hallmark in numerous models of myelin-related disorders (*Patzig et al., 2016a*) and upon normal brain aging (*Peters, 2002*; *Sturrock, 1976*). Localized in the non-compacted adaxonal myelin layer, i.e. underlying the innermost compact myelin membrane, septin filaments scaffold the myelin structure, thereby preventing the emergence of myelin outfoldings (*Patzig et al., 2016a*). The phenotype of *Anln^flox/flox;Cnp^Cre/WT* mice is very similar to that of mice lacking SEPT8, a septin monomer essential for the assembly of myelin septin filaments (*Patzig et al., 2016a*). Thus, the present study has revealed that anillin is critical for the assembly of septin filaments in CNS myelin and that the lack of these filaments causes myelin outfoldings associated with reduced nerve conduction velocity.

During yeast cytokinesis, PtdIns(4,5)P$_2$ recruits the anillin homolog Mid2p to the membrane at the cleavage furrow, which in turn is critical for the polymerization of septin subunits into rings at the budding site of mother and daughter cells (*Bertin et al., 2010*; *Liu et al., 2012*). At the molecular level, the interactions between PtdIns(4,5)P$_2$, anillin and septins are probably principally conserved between dividing cells and mature oligodendrocytes, which are post-mitotic. Yet, during cytokinesis, ANLN affects microtubules and myosin-dependent bundling of actin filaments (*Hickson and O'Farrell, 2008a*; *Hickson and O'Farrell, 2008b*). Since myelin wrapping involves actin disassembly (*Nawaz et al., 2015*; *Zuchero et al., 2015*) and myelin compaction requires trafficking of *Mbp* mRNA along microtubules (*Müller et al., 2013*; *Thakurela et al., 2016*), we point out that there is no evidence of fewer, thinner or non-compacted myelin sheaths in *Anln^flox/flox;Cnp^Cre/WT* mice. Instead, ANLN is required during the latest stages of myelin maturation, i.e., for septin-dependent scaffolding of the myelin sheath rather than for its actin/tubulin-dependent biosynthesis and compaction.

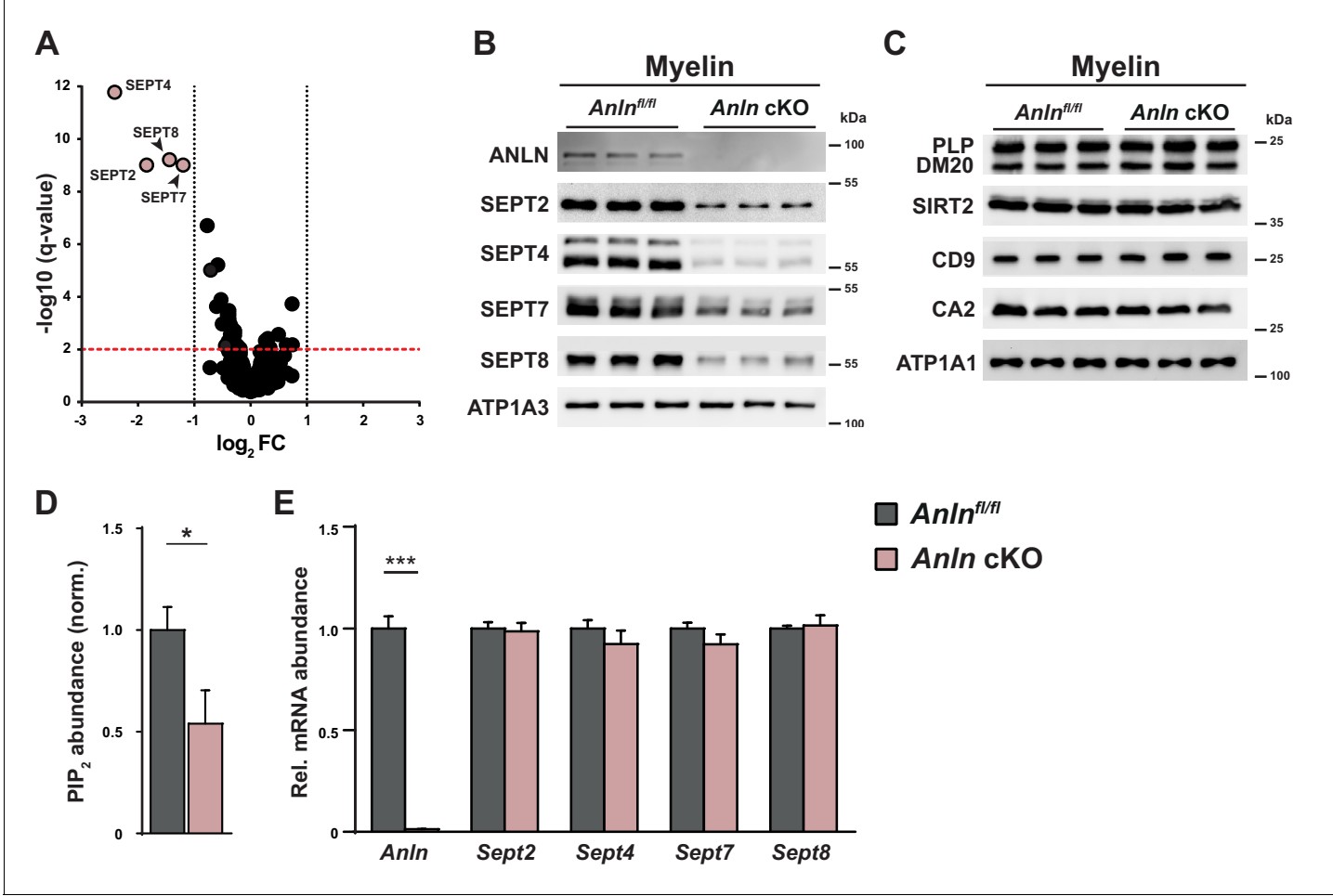

**Figure 3.** Myelin composition in mice lacking oligodendroglial expression of ANLN. (**A**) Volcano plot summarizing genotype-dependent quantitative myelin proteome analysis. Data points represent quantified proteins in myelin purified at P75 from the brains of *Anln* cKO compared to *Anln*<sup>fl/fl</sup> mice (n = 3 mice per genotype). Data points are plotted as log2-transformed fold-change (FC) on the x-axis against the −log10-transformed q-value on the y-axis. The horizontal red dashed line indicates a q-value of q = 0.01; the vertical black dashed lines mark the ±1 log2 fold-change threshold indicating a halved or doubled abundance of a protein in myelin, respectively. Data points representing myelin septin monomers (SEPT2, SEPT4, SEPT7, SEPT8) are highlighted in light red color with protein names given; note that their abundance is strongly reduced in *Anln* cKO compared to *Anln*<sup>fl/fl</sup> myelin. Also note that ANLN is not represented because it was not detected in *Anln* cKO myelin. For bar graphs showing genotype-dependent comparison of the abundance of individual proteins in myelin see *Figure 3—figure supplement 1A–D*. For the original dataset and exact q-values see *Figure 3—source data 1*. (**B**) Immunoblotting validates the lack of anillin (ANLN) and the strong reduction of septins (SEPT2, SEPT4, SEPT7, SEPT8) in myelin purified from the brains of *Anln* cKO-mice. ATPase Na+/K + transporting subunit alpha 3 (ATP1A3) was detected as control. Blot shows n = 3 mice per genotype. (**C**) Immunoblotting indicates that the abundance of classical myelin proteins (PLP/DM20, SIRT2, CD9, CA2) is unaltered in myelin purified from the brains of *Anln* cKO-mice. ATP1A1 served as control. Blot shows n = 3 mice per genotype. (**D**) Genotype-dependent quantitative assessment of PtdIns(4,5)$P_2$ (PIP$_2$)–levels in myelin purified from the brains of *Anln* cKO-mice compared to controls (*Anln*<sup>fl/fl</sup>) at P75. Mean +/SEM. n = 6 mice per genotype; two-tailed unpaired t-test; PtdIns(4,5)P$_2$p=0.0435. (**E**) qRT-PCR to determine the abundance of mRNAs encoding anillin and myelin septins in the white matter (corpus callosum) of control (*Anln*<sup>fl/fl</sup>) versus *Anln* cKO-mice. Note that *Anln* mRNA was virtually undetectable in *Anln* cKO-mice while the abundances of *Sept2*, *Sept4*, *Sept7* and *Sept8* mRNAs were unaltered. Mean +/SEM. n = 6 mice per genotype; two-way ANOVA; *Anln* p<0.0001, *Sept2* p>0.9999, *Sept4* p>0.9999, *Sept7* p>0.9999, *Sept8* p>0.9999.
DOI: https://doi.org/10.7554/eLife.43888.008

The following source data and figure supplement are available for figure 3:

**Source data 1.** Label-free quantification of proteins in myelin purified from the brains of Anln cKO and control mice Tryptic peptides derived from two technical replicates (replicate digestion) per biological replicate (n = 3 mice per genotype) were analyzed by LC-MS (12 runs in total).
DOI: https://doi.org/10.7554/eLife.43888.010

**Figure supplement 1.** Genotype-dependent quantification of selected proteins according to myelin proteome analysis.
DOI: https://doi.org/10.7554/eLife.43888.009

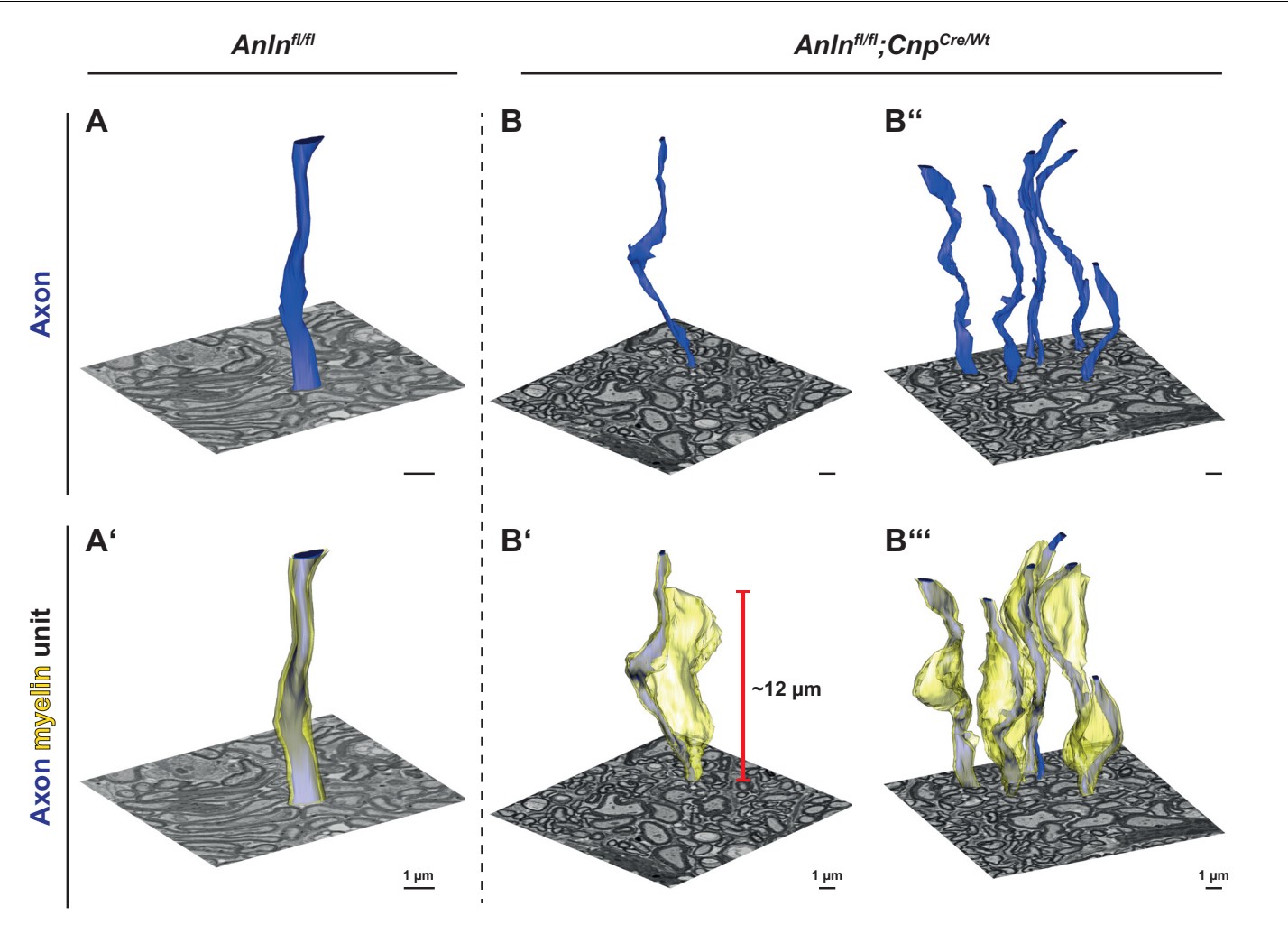

**Figure 4.** Three-dimensional reconstruction of myelin outfoldings in Anln cKO mice. (**A–C**) Focused ion beam-scanning electron microscopy (FIB-SEM) micrographs and 3D reconstruction of the plasma membrane of myelinated axonal segments (blue) and respective myelin sheaths (yellow) of representative axon/myelin-units in the optic nerve of control (*Anln*$^{fl/fl}$) (**A,A'**) and *Anln* cKO (**B,B',B'',B'''**) mice at 5.5 mo. Note the tight association of the myelin sheath reconstructed in **A'** with the corresponding axon (in **A,A'**) over at least 10 μm in the control nerve. An individual myelin outfolding (**B'**) and the corresponding axon are reconstructed over 20 μm in B,B'. All myelin outfoldings in that same block (as in **B,B'**) were reconstructed in B''' with their corresponding axons (in **B'',B'''**). Note that myelin outfoldings represent large sheets of compacted multilayered membrane stacks that extend considerably away from their respective myelinated axon, commonly displaying longitudinal dimensions between 10 μm and 15 μm. See *Videos 1–3*.
DOI: https://doi.org/10.7554/eLife.43888.011

Myelin is one of the most long-lived structures in the CNS (*Toyama et al., 2013*) with a particularly slow turnover rate of its components (*Lasiene et al., 2009*; *Lüders et al., 2019*; *Yeung et al., 2014*; *Young et al., 2013*). Failure to physically stabilize the myelin architecture causes myelin outfoldings and affects nerve conduction velocity. We therefore propose that PtdIns(4,5)P$_2$/anillin-dependent scaffolding of myelin by septin filaments represents a crucial step of myelin maturation.

## Materials and methods

### Mouse models

Embryonic stem cells (ES) harboring an engineered allele of the *Anln* gene were acquired from the European Conditional Mouse Mutagenesis Program (Eucomm). ES were microinjected into blastocysts derived from c57BL/6N mice, and embryos were transferred to pseudo-pregnant foster

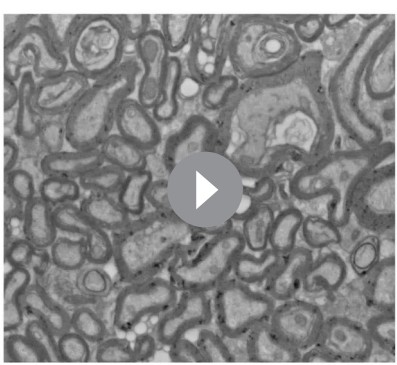

**Video 1.** FIB-SEM and 3D reconstruction of a normal-appearing axon/myelin-unit in a control mouse.
DOI: https://doi.org/10.7554/eLife.43888.012

mothers, yielding two chimeric males. For ES clone EPD0545_1_F09, germline transmission was achieved upon breeding with c57BL/6N-females, yielding mice harboring the *Anln*[LacZ-neo] allele. The lacZ-neo cassette was excised in vivo upon interbreeding with mice expressing FLIP recombinase (*129S4/SvJaeSor-Gt(ROSA)26Sor*[tm1 (FLP1)Dym/J]; backcrossed into c57BL/6N), yielding mice carrying the *Anln*[flox] allele. To inactivate expression of ANLN in myelinating cells, exon four was excised in vivo upon appropriate inter-breedings of *Anln*[flox] mice with mice expressing Cre recombinase under control of the *Cnp* promoter (*Lappe-Siefke et al., 2003*). For simplicity, *Anln*[fl/fl];*Cnp*[Cre/WT] mice are also termed *Anln* conditional knockout (*Anln* cKO). Routine genotyping of the *Anln* allele as shown in *Figure 2—figure supplement 1B* was performed by PCR with sense primer P1 (5'-GACATAGCCC TCAGTGTT-CAGG; binding 5 'of the first loxP-site) in combination with antisense primers P2 (5'-GAATCCTGCA TGGACAGACAG; binding the segment flanked by loxP-sites), and P3 (5'-GAGCT-CAGAC CATAACTTCG; binding 3 'of the third loxP site). PCR genotyping of the *Cnp* allele was with primers *2016* (5'-GCCTTCAAAC TGTCCATCTC), *7315* (5'-CCCAGCCCTT TTATTACCAC), *4193* (5'-CCTGGAAAAT GCTTCTGTCCG) and *4192* (5'-CAGGGTGTTA TAAGCAATCCC). Experimental mutant mice were analyzed together with littermate controls as far as possible. Mice were kept in the mouse facility of the Max Planck Institute of Experimental Medicine with a 12 hr light/dark cycle and 2–5 mice per cage. All experiments were approved by the Niedersächsisches Landesamt für Verbraucherschutz und Lebensmittelsicherheit (license 33.19-42502-04-15/1833) in agreement with the German Animal Protection Law.

## Quantifications and statistical analysis

Sample size was according to previous analyses of similar parameters, e.g. in (*Patzig et al., 2016a*). All quantifications were performed blinded with respect to the genotypes. Bar graphs display mean values and standard error of the mean (SEM). Statistical tests were performed in GraphPad Prism

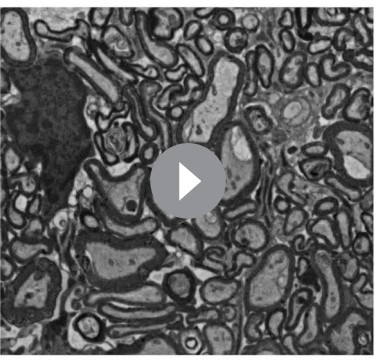

**Video 2.** FIB-SEM and 3D reconstruction of one selected myelin outfolding in an *Anln* cKO mouse.
DOI: https://doi.org/10.7554/eLife.43888.013

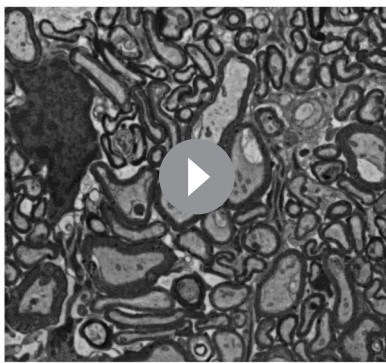

**Video 3.** FIB-SEM and 3D reconstruction of multiple myelin outfoldings in one tissue block of an *Anln* cKO mouse.
DOI: https://doi.org/10.7554/eLife.43888.014

6.0. Tests were chosen depending on experimental groups and as suggested by the software. To test for variance, F-test was performed in GraphPad Prism 6.0. GraphPad online test at http://graph-pad.com/quickcalcs/Grubbs1.cfm was used to test for outliers; however, no outliers were removed from the data. Levels of significance were set at $p<0.05$ (*), $p<0.01$ (**), and $p<0.001$ (***). Exact p-values are given in the figure legends. For myelin proteome analysis, q-values were calculated by R data analysis as detailed in the section 'Myelin proteome analysis' and given in *Figure 3—source data 1*.

## Electron microscopy

For conventional transmission electron microscopy, sample preparation by chemical fixation or by high pressure freezing and freeze substitution was performed as described (*Möbius et al., 2010*; *Möbius et al., 2016*; *Patzig et al., 2016a*; *Patzig et al., 2016b*). Myelinated and degenerated axons were assessed on electron micrographs of the optic nerves of 4–5 male mice per genotype chemically fixed at 6 mo of age. 15 randomly distributed non-overlapping electron micrographs were taken per optic nerve at 7000x magnification (one field = 220 µm$^2$). Electron micrographs were assessed using ImageJ (Fiji) (*Schindelin et al., 2012*). A minimum of 1600 axons per mouse was assigned to one of three categories: healthy-appearing myelinated axons, healthy-appearing non-myelinated axons and degenerating/degenerated profiles. Axons were counted as myelinated if ensheathed by at least one complete layer of compact myelin. Degenerating/degenerated profiles were identified by the presence of tubovesicular structures and amorphous cytoplasm within an axon or the absence of an identifiable axon within a myelin sheath, respectively. The area occupied by myelin outfoldings was assessed by applying a point-hit counting method (*Edgar et al., 2009*; *Patzig et al., 2016a*) to the same electron micrographs. Briefly, a regular grid of 0.25 µm$^2$ was placed on the images. The number of intercepts coinciding with myelin outfoldings was related to the evaluated area. g-ratios were calculated as ratio between axonal Feret diameter and Feret diameter of the corresponding myelin sheath. To this aim, the top left quarter of the same electron micrographs (one field = 55 µm$^2$) was assessed, yielding a minimum of 200 myelinated axons per mouse.

For focused ion beam-scanning electron microscopy (FIB-SEM), optic nerves dissected from mice at 5.5 months of age were fixed for 24 hr in 4% formaldehyde (Serva) and 2.5% glutaraldehyde (Science Services) in 0.1 M phosphate buffer (PB). The samples were processed principally following the OTO protocol (www.ncmir.ucsd.edu/sbem-protocol) (*Deerinck et al., 2010*) with some modifications: Samples were washed in 0.1 M PB (3 × 15 min), incubated for 3 hr at 4°C in 2% osmium tetroxide (OsO$_4$) (Electron Microscopy Sciences) and 0.25% potassium ferrocyanide (K$_4$[Fe(CN)$_6$]) (Electron Microscopy Sciences), washed with H$_2$O (3 × 15 min) and then incubated with 0.1% thiocarbohydrazide (Sigma-Aldrich) for 1 hr at room temperature. For further contrast enhancement the tissue was treated with 2% OsO$_4$ for 90 min at room temperature. The samples were then washed with H$_2$O (3 × 15 min), contrasted overnight at 4°C with 2% uranyl acetate (SPI-Chem) and washed again with H$_2$O (3 × 15 min), followed by dehydration in an increasing acetone series (30%, 50%, 75%, 90%, 3 × 100%). The tissue was infiltrated with increasing concentrations of Durcupan (Sigma-Aldrich, components A, B, C) for 2 hr each (25%, 50%, 75% Durcupan in acetone) and then incubated in 100% Durcupan overnight. Fresh Durcupan with accelerator (component D) was added to the samples for 5 hr before embedding the samples in resin blocks. The blocks were polymerized for 48 hr at 60°C.

The blocks were trimmed with a 90° diamond trimming knife (Diatome AG, Biel, Switzerland). The blocks were then attached to the SEM stub (Science Services GmbH, Pin 12.7 mm x 3.1 mm) by a silver filled epoxy resin (Epoxy Conductive Adhesive, EPO-TEK EE 129–4; EMS) and polymerized at 60° overnight. The samples were coated with a 10 nm platinum layer using the sputter coater EM ACE600 (Leica) at 35 mA current. Samples were placed into the Crossbeam 540 focused ion beam-scanning electron microscope (Carl Zeiss Microscopy GmbH). To ensure even milling and to protect the surface, a 400 nm platinum layer was deposited on top of the region of interest. Atlas 3D (Atlas 5.1, Fibics, Canada) software was used to collect the 3D data. Samples were exposed with a 15 nA current, and a 7 nA current was used to polish the surface. The images were acquired at 1.5 kV with the ESB detector (450 V ESB grid, pixel size x/y 2 nm) in a continuous mill-and-acquire mode using 700 pA for the milling aperture (z-step 50 nm).

For image analysis, alignments were done with TrackEM2 (*Cardona et al., 2012*), a plugin of Fiji (*Schindelin et al., 2012*). The following post-processing steps were performed in Fiji: The dataset was cropped and inverted before applying a Gaussian blurr (sigma 2) and local contrast

enhancement (CLAHE: blocksize 56; histogram bins 100; maximum slope 1.5). The images were manually segmented using IMOD (*Kremer et al., 1996*).

## Immunohistochemistry

Immunohistochemistry on sections of paraffin-embedded brains to determine neuropathology was performed as described (*de Monasterio-Schrader et al., 2013*; *Patzig et al., 2016a*), assessing five male mice per genotype at postnatal day 75 (P75). Antibodies were specific for MAC3 (Pharmingen 553322; 1:400), glial fibrillary acidic protein (GFAP) (NovoCastra NCL-L-GFAP-GA5; 1:200) or amyloid precursor protein (APP) (Chemicon MAB348; 1:1000). For quantification, the hippocampal fimbria was selected, and APP-positive axonal spheroids were counted. Microscopy was as described (*Patzig et al., 2016a*). To quantify white matter area immunopositive for MAC3 or GFAP, the hippocampal fimbria was selected on micrographs and analyzed using an ImageJ plugin for semiautomated analysis (*de Monasterio-Schrader et al., 2013*; *Lüders et al., 2017*; *Patzig et al., 2016a*). Data were related to the mean of wild-type levels. Silver impregnation of myelin on histological sections was as described (*Gallyas, 1979*; *Patzig et al., 2016a*). Microscopy and image stitching was as described (*Patzig et al., 2016a*).

Immunohistochemistry on cryosectioned optic nerves to assess expression and localization of ANLN and myelin septins was as described (*Patzig et al., 2016a*). Antibodies were specific for ANLN (Acris AP16165PU-N; 1:200), SEPT7 (IBL18991; 1:1000), SEPT8 (ProteinTech Group 11769–1-AP; 1:500), TUJ1 (Covance MMS-435P; 1:1000), neurofilament (Covance SMI-31; 1:1500), myelin-associated glycoprotein (MAG clone 513; Chemicon MAB1567; 1:50), voltage-gated sodium channel $Na_v1,6$ (alomonelabs ASC-009; 1:500) or contactin-associated protein (CASPR; Neuromabs 75–001; 1:500). Secondary antibodies were donkey $\alpha$-rabbit-Alexa488 (Invitrogen A21206), donkey $\alpha$-mouse-Alexa488 (Invitrogen A21202), donkey $\alpha$-rabbit-Alexa555 (Invitrogen A31572), donkey $\alpha$-mouse-Alexa555 (Invitrogen A31570), donkey $\alpha$-goat-Cy3 (dianova 705-165-147) and donkey $\alpha$-mouse Dye-light633 (Yo-Pro). Images were obtained by confocal microscopy (Leica SP5) as described (*Patzig et al., 2016a*). The LAS AF lite and Fiji were used to export the images as tif-files. Imaris was used for 3D-reconstructions. For quantification of nodal density, the frequency of occurrence of two CASPR-immunopositive paranodes was analyzed using Fiji. CASPR-immunopositivity was converted using a threshold and counted using ITNC plugin (n = 4 mice per genotype, one section each, five random micrographs of spinal cord white matter with a size of 2500 $\mu m^2$ per micrograph). Statistical analysis was performed using GraphPad Prism 6.0.

## Myelin purification

A light-weight membrane fraction enriched for myelin was purified from mouse brains by sucrose density centrifugation and osmotic shocks as described (*Jahn et al., 2013*; *Patzig et al., 2016a*). For immunoblot analyses of myelin during development, male wild-type (c57Bl/6N) mice were used at the indicated ages. For proteome and immunoblot analyses of *Anln* cKO mice and control (*Anln^{fl/fl}*) littermates, n = 3 male mice at P75 were used. Protein concentrations were determined using the DC protein assay (BioRad). For PtdIns(4,5)P$_2$ measurement, myelin was purified from n = 6 male mice per genotype at P75 with phosphatase inhibitor (Roche PhosSTOP; 1 tablet per 10 ml) added to the Tris-buffered saline and the sucrose solutions.

## Myelin proteome analysis

Differential quantitative label-free proteome analysis of myelin purified from the brains of male *Anln* cKO mice and control littermates at P75 was performed using a label-free quantification workflow essentially as described (*Ambrozkiewicz et al., 2018*; *Patzig et al., 2016a*). Briefly, protein fractions corresponding to 10 $\mu g$ myelin protein were lysed and reduced in lysis buffer (7 M urea, 2 M thiourea, 10 mM DTT, 0.1 M Tris pH 8.5) containing 1% ASB-14 while shaking for 30 min at 37°C. Subsequently, samples were diluted with 10 volumes lysis buffer containing 2% CHAPS to reduce the ASB-14 concentration and processed according to an automated filter-aided sample preparation (FASP) protocol for in-solution digestion with trypsin. Aliquots of the recovered tryptic peptides were spiked with 10 fmol/$\mu l$ Hi3 EColi standard (Waters Corporation) for protein quantification according to the TOP3 approach (*Silva et al., 2006*). This standard contains a set of quantified synthetic peptides representing the top six ionizing tryptic peptides derived from E. coli. Chaperone protein ClpB.

Peptide samples were directly subjected to analysis by liquid chromatography coupled to electro-spray mass spectrometry (LC-MS) on a Synapt G2-S quadrupole time-of-flight mass spectrometer equipped with ion mobility option (Waters Corporation). Analyses were performed in the ion mobility-enhanced data-independent acquisition mode with drift time-specific collision energies as described (*Distler et al., 2014*; *Distler et al., 2016*). Specifically, a novel data acquisition strategy with dynamic range enhancement (DRE) was used, in which a deflection lens cycles between full and reduced ion transmission during one scan. This method provides an optimal trade-off between identification rate (i.e. proteome depth) and dynamic range for correct quantification of high-abundant myelin proteins. Continuum LC-MS data were processed using Waters ProteinLynx Global Server (PLGS) version 3.0.2 and database searches were performed against the UniProtKB/Swiss-Prot mouse proteome (release 2016–07, 16806 entries) to which the sequence information for E. coli. Chaperone protein ClpB, porcine trypsin, and the reversed sequence of each entry was added. The false discovery rate (FDR) for protein identification was set to 1% threshold. As to the experimental design, myelin protein fractions from the CNS of three mice per condition (*Anln* cKO, Ctrl) were processed with replicate digestion, resulting in two technical replicates per biological replicate and thus in a total of 12 LC-MS runs to be compared in the freely available software ISOQuant (www.iso-quant.net). This post-identification analysis included retention time alignment, exact mass and retention time (EMRT) and ion mobility clustering, data normalization, isoform/homology filtering, and calculation of absolute in-sample amounts for each detected protein as described (*Ambrozkiewicz et al., 2018*; *Kuharev et al., 2015*). FDR for both peptides and proteins was set to 1% threshold and only proteins reported by at least two peptides were quantified using the TOP3 method. The parts per million (ppm) abundance values (i.e. the relative amount (w/w) of each protein in respect to the sum over all detected proteins) were log2-transformed and significant changes in protein abundance were detected by moderated t-statistics with an empirical Bayes approach and false discovery (FDR)-based correction for multiple comparisons performed in RStudio (*Ambrozkiewicz et al., 2018*; *Kammers et al., 2015*). The genotype-dependent relative abundance of a protein in myelin was compared with high stringency and accepted as altered if both statistically significant (q-value <0.01) and exceeding a regulation factor threshold of 2-fold.

## Lipid extraction and PtdIns(4,5)P$_2$ measurement

Purified myelin was thawed on ice in 1 ml of an acidic extraction solvent (*Cho and Boss, 1995*) containing 36% (v/v) $CH_3OH$, 36% (v/v) $CHCl_3$, 18% (v/v) 2.4 M HCl, and 9% (v/v) 0.4 M EDTA in a glass reaction vial (73750–13100, Kimble-Chase, Meiningen, Germany). The material was ground to homogeneity using a rotating Douncer (IKA, Staufen, Germany) on ice. Samples were mixed and incubated for 2 hr at 4°C while shaking on a Cat-Ing shaker (Ballrechten, Germany, city). Phases were separated by centrifugation for 2 min at 600 *g*, and the organic phase was collected into a fresh glass tube. Samples were re-extracted twice with 500 µl of $CHCl_3$. The combined organic phases were washed twice using 1.5 ml of 0.5 M HCl in 50% (v/v) $CH_3OH$. The first aqueous phase was discarded; after the second washing step the organic phase was collected into a fresh glass tube. The lipid extracts were analyzed in a double-blind experiment for phosphatidylinositol (4,5)-bisphosphate (PtdIns(4,5)P$_2$) using combined thin layer chromatography (TLC) and gas chromatography (GC) essentially as previously described (*Goebbels et al., 2010*; *König et al., 2008*). Briefly, myelin samples were subjected to TLC on silica S60 plates (Merck, Darmstadt, Germany) using a developing solvent of $CHCl_3:CH_3OH:NH_4OH:H_2O$ (57:50:4:11 v/v/v/v) (*Perera et al., 2005*). Lipids were identified by co-migration with authentic standards (5 µg; Avanti Polar Lipids, Alabaster, AL, USA), re-isolated and quantified according to their fatty acid content, as determined by GC. For GC analysis, re-isolated lipids were dissolved in MeOH/toluol (2:1 v/v), 5 µg of tripentadecanoin was added as an internal standard for quantification and the mixture transmethylated with 0.5 M sodium methoxide (Sigma-Aldrich, Munich, Germany) according to (*Hornung et al., 2002*). After 30 min incubation at room temperature, derivatization was terminated by adding 0.5 ml 5 M NaCl (Sigma-Aldrich, Munich, Germany) and 50 µl of 32% HCl (Carl Roth, Karlsruhe, Germany). Fatty acid methyl esters were extracted with 2 ml hexane (Carl Roth, Karlsruhe, Germany), the hexane phase was washed twice with 2 ml dd$H_2O$ and dried under streaming nitrogen. The lipid coat was resuspended in 10 µl acetonitrile (Carl-Roth, Karlsruhe, Germany) and transferred to GC vials (order number 702287.1, VWR, Darmstadt, Germany). GC analysis was performed using a GC2010plus gas chromatograph with flame ionization detection (Shimadzu, Jena, Germany) fitted with a DB-23 capillary column (30

m x 250 µm, 0.25 µm coating thickness; J and W, Agilent, Waldbronn, Germany). Helium flowed as a carrier gas at 1 ml min$^{-1}$. Samples were injected at 220°C. The temperature gradient was 150°C for 1 min, 150–200°C at 8 °C min$^{-1}$, 200–250°C at 25 °C min$^{-1}$ and 250°C for 6 min as previously described (König et al., 2008). Fatty acids were identified according to authentic standards and quantified relative to the internal standard using GC-solution software (Shimadzu, Jena, Germany).

## Immunoblotting

Immunoblotting was performed as described (Patzig et al., 2016a; Schardt et al., 2009). Antibodies were specific for ANLN (Acris AP16165PU-N; 1:1000), SEPT2 (ProteinTech Group 11397–1-AP; 1:500), SEPT4 (IBL JP18987; 1:500), SEPT7 (IBL JP18991; 1:5000), SEPT8 (ProteinTech Group 11769–1-AP; 1:2500), MAG ((Erb et al., 2003); kindly provided by N. Schaeren-Wiemers, Basel; 1:500), PLP/DM20 (A431 ((Jung et al., 1996); 1:5000), cyclic nucleotide phosphodiesterase (CNP) (Sigma C5922; 1:1000), MOG (clone 8–18 C5 (Linnington et al., 1984); 1:5000; kindly provided by C. Linington, Glasgow), SIRT2 (abcam 67299; 1:500), CD9 (abcam ab92726; 1:2000), CA2 ((Ghandour et al., 1980a; Ghandour et al., 1980b); 1:1000; kindly provided by S. Ghandour, Strasbourg), ATP1A1 (abcam ab7671; 1:2500), ATP1A3 (abcam ab2826; 1:1000), beta3-Tubulin (TUBB3/Tuj1) (Covance MMS-435P; 1:1000). Secondary HRP-coupled antibodies were anti-mouse (dianova 715-035-020; 1:1000), -rabbit (dianova 711-035-152; 1:1000), or -goat (dianova 705-035-003; 1:500). Immunoblots were scanned using the Intas ChemoCam system.

## Quantitative RT-PCR

mRNA abundance was determined by qRT-PCR as described (Lüders et al., 2017; Patzig et al., 2016a) using corpus callosi dissected from 4 months old male and female mice of the indicated genotypes. mRNA abundance was analyzed in relation to the mean of the standard Ppia, which did not differ between genotypes. Statistical analysis was performed in GraphPad Prism 6.0. Primers were specific for Anln (forward 5'-ACAATCCAAG GACAAACTTGC, reverse 5'- GCGTTCCAGG AAAGGCTTA, Sept2 (forward 5'-TCCTGACTGA TCTCTACCCAGAA, reverse 5'-AAGCCTCTAT CTGGACAGTTCTTT), Sept4 (forward 5'-ACTGACTTGT ACCGGGATCG, reverse 5'-TCTCCACGGT TTGCATGAT), Sept7 (forward 5'-AGAGGAAGGC AGTATCCTTGG, reverse 5'-TTTCAAGTCC TGCATATGTGTTC), Sept8 (forward 5'-CTGAGCCCCG GAGCCTGT, reverse 5'-CAATCCCAGT TTCGCCCACA), Cdc42 (forward 5'-GCTGTCAAGT ATGTGGAGTGCT, reverse 5'-GGCTCTTCTT CGGTTCTGG), Rhob (forward 5'-CAGACTGCCT GACATCTGCT, reverse 5'-GTGCCCACGCT AATTCTCAG) and the standard Ppia (forward 5'-CACAAACGGT TCCCAGTTTT, reverse 5'-TTCCCAAAGA CCACATGCTT).

## Nerve conduction velocity measurement

Nerve conduction velocity in the CNS was measured in 6 months old anesthetized male mice in vivo as described (Dibaj et al., 2012; Patzig et al., 2016a).

## Acknowledgements

We thank A Fahrenholz, D Hesse, U Kutzke, C Nardis, T Ruhwedel and M Schindler for technical support, S Ghandour, C Linington and N Schaeren-Wiemers for antibodies, and S Tenzer for discussions. AMS was funded by the Cluster of Excellence and Deutsche Forschungsgemeinschaft (DFG) Research Center Nanoscale Microscopy and Molecular Physiology of the Brain (CNMPB). K-AN holds an ERC Advanced Grant ('MyeliNano'). This work is supported by the Deutsche Forschungsgemeinschaft (DFG) (Grants WE 2720/2–2 and WE 2720/4–1 to HBW).

## Additional information

### Competing interests

Klaus-Armin Nave: Reviewing editor, eLife. The other authors declare that no competing interests exist.

## Funding

| Funder | Grant reference number | Author |
|---|---|---|
| Deutsche Forschungsgemeinschaft | WE2720/2-2 | Hauke B Werner |
| Deutsche Forschungsgemeinschaft | WE2720/4-1 | Hauke B Werner |
| European Research Council | Advanced Grant MyeliNano | Klaus-Armin Nave |

The funders had no role in study design, data collection and interpretation, or the decision to submit the work for publication.

## Author contributions

Michelle S Erwig, Investigation, Performed all experiments not specified below, conducted statistical analysis, contributed to analysis and interpretation of data, contributed to writing the manuscript and approved the version to be published; Julia Patzig, Investigation, Contributed to conception and design of the study, contributed to writing the article and approved the version to be published; Anna M Steyer, Investigation, Performed, analyzed and interpreted the FIB-SEM, contributed to writing the article and approved the version to be published; Payam Dibaj, Investigation, Performed, analyzed and interpreted the electrophysiological measurements, contributed to writing the article and approved the version to be published; Mareike Heilmann, Investigation, Performed and analyzed the PIP2 measurement, contributed to writing the article and approved the version to be published; Ingo Heilmann, Supervision, Methodology, Analyzed and interpreted the PIP2 measurement, contributed to writing the article and approved the version to be published; Ramona B Jung, Investigation, Contributed to biochemical analysis, contributed to writing the article and approved the version to be published; Kathrin Kusch, Supervision, Investigation, Performed, analyzed and interpreted the qRT-PCR, contributed to writing the article and approved the version to be published; Wiebke Möbius, Supervision, Methodology, Contributed to analyzing and interpreting electron microscopy and FIB-SEM data, contributed to writing the article and approved the version to be published; Olaf Jahn, Investigation, Methodology, Performed, analyzed and interpreted the proteome analysis, contributed to writing the article and approved the version to be published; Klaus-Armin Nave, Writing—review and editing, Contributed to designing experiments, contributed to writing the article and approved the version to be published; Hauke B Werner, Conceptualization, Supervision, Writing—original draft, Writing—review and editing, Conceived, designed and directed the study, analyzed and interpreted data, wrote the article and approved the version to be published

## Author ORCIDs

Wiebke Möbius (iD) http://orcid.org/0000-0002-2902-7165
Klaus-Armin Nave (iD) http://orcid.org/0000-0001-8724-9666
Hauke B Werner (iD) http://orcid.org/0000-0002-7710-5738

## Ethics

Animal experimentation: All experiments were approved by the Niedersächsisches Landesamt für Verbraucherschutz und Lebensmittelsicherheit (license 33.19-42502-04-15/1833) in agreement with the German Animal Protection Law.

## Decision letter and Author response

Decision letter https://doi.org/10.7554/eLife.43888.018

# Additional files

## Supplementary files

• Transparent reporting form
DOI: https://doi.org/10.7554/eLife.43888.015

**Data availability**

All data are included in the manuscript and supporting files. A source data file is provided for Figure 3.

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
