## [Decision Letter]

[Editors’ note: minor issues and corrections have not been included, so there is not an accompanying Author response.]

Congratulations, we are pleased to inform you that your article, "Anillin facilitates septin assembly to prevent pathological outfoldings of central nervous system myelin", has been accepted for publication in *eLife*.

This study represents an important advance on their previous study from these authors, identifying septins as crucial to the maintenance of mature myelin morphology in the CNS, and seems entirely appropriate for publication in *eLife* as a 'Research Advance' paper. Here, the authors assess the role of the cytoskeletal adaptor anillin in CNS myelination: they convincingly show that anillin is expressed and localized similarly to septin in myelinating glia, and that its disruption leads to very specific effects on myelin maintenance. The authors further demonstrate that anillin appears essential to the localization of septin proteins to myelin itself, which is a key observation central to the premise that anillin is a key regulator of septin assembly for myelin maintenance. Specific reviewer comments are included below.

*Reviewer #1:*

"Anillin facilitates septin assembly to prevent pathological outfoldings of central nervous system myelin", by Erwig et al., represents an important advance on their previous study identifying septins as crucial to the maintenance of mature myelin morphology in the central nervous system, and seems entirely appropriate for publication in *eLife*. In their new study, Erwig et al. assess the role of the cytoskeletal adaptor anillin in CNS myelination: they find that it is expressed and localised similarly to septin in myelinating glia, and that its disruption leads to very specific effects on myelin maintenance, i.e. the formation of myelin outfoldings. The authors further demonstrate that anillin appears essential to the localisation of septin proteins to myelin itself, which is a key observation central to the premise that anillin is a key regulator of septin assembly for myelin maintenance.

The experiments incorporate the generation and analyses of conditional knockout mice, biochemistry, mass spec analyses of myelin, robust assessments of protein localisation over time, detailed electron microscopic assessments of myelination in mutants, electrophysiological assessments of function, and 3D electron microscopy-based reconstructions of outfolding pathologies. The core analyses are carried out to the absolute highest standard, the presentation is clear, and the conclusions drawn entirely in line with the data, and further supported by additional controls, e.g. that dysregulation of nodes of Ranvier may contribute to conduction impairments.

I don't actually have any major criticisms at all. One can always ask for more analyses and deeper mechanistic probing, but to me this manuscript already provides an important advance and could be published as it is.

*Reviewer #2:*

This group has previously suggested in a beautiful paper published in *eLife* that anillin and septins form filaments along the myelin unit, which serves to maintain the normal compact organization of the myelin internode. They also convincingly showed that the absence of one of the septins results in aberrant sheath extension and the formation of myelin outfolding. In the current work, they examine the role of the adapter protein anillin which associates with septins and show that genetic deletion of the corresponding gene results in a similar neuropathology to mice lacking Sept8 (i.e., about 3% of the fibers exhibit myelin outfolding). They also found that the absence of anillin causes reduced expression of several cytoskeletal proteins including septins, as well as reduced level of phosphoinositides. They conclude that anillin is required for a septin-based scaffold of the myelin sheath.

*Reviewer #3:*

This study by Erwig et al. focuses on the impact of septin assembly in central nervous system myelin. In an earlier *eLife* paper (Patzig et al., 2016), this group demonstrated that the loss of septins in myelin results in outfoldings of myelin, and that a septin-interacting protein, anillin, was also downregulated. This manuscript takes that observation further, demonstrating that conditional deletion of anillin results in essentially the same phenotype as the septin conditional deletion animals.

The data in this manuscript are well developed and convincing. The writing is straightforward. However, several points raise concerns as to whether this study moves the field forward significantly. The interaction of septins and anillin has been established for at least 15 years, and in many papers the concept that anillin "instructs" septin filament assembly is presented. Thus, the observation that loss of anillin acts comparably to that of septins is not unexpected. The involvement of anillin in septin filament assembly in myelin was clearly established in the earlier Patzig et al. paper, where many virtually identical analyses were conducted, demonstrating, for example, that the loss of septins induced loss of anillin protein but not RNA, and here, the loss of anillin induces loss of septin proteins but not RNA. This study clearly presents new information and as noted above the studies were executed well. Nevertheless, the new data primarily confirm published work showing that anillin is involved in septin organization in a number of cells, and the earlier Patzig study indicating that septin assembly including the involvement of anillin is crucial for myelin structure, i.e. it prevents the inappropriate outfolding of compacted myelin.

One element of this and the earlier Patzig study that is unexpected is that conduction velocity is altered at P75 in both types of septin/anillin knockouts but there is no obvious axonal pathology. Nevertheless, in both sets of animals the myelin outfoldings dramatically increase as the animals age. One unaddressed question is whether there is axonal pathology or greater conduction velocity change in 6M or older animals when myelin outfoldings increase.

Overall, this is a solid study, extending the earlier work of this group focused on the role of the septin assembly in myelin structure. However, the myelin outfolding outcome in mice conditionally deleted for anillin is not unexpected, and no significantly greater insight is provided by this study, with respect to understanding how septin assemblies regulate and prevent such outfoldings.